# Clinico-pathological and treatment characteristics of HIV and non-HIV related vulvar cancers: Analysis of a South African cohort

Adekunle Emmanuel Sajo[1]*, Edwin Francis Mnisi[1,2], Sheynaz Bassa[3,4], Cathy Visser[1], Greta Dreyer[1,5]

1 Department of Obstetrics and Gynaecology, Gynaecologic Oncology Unit, University of Pretoria, Pretoria, South Africa, 2 Department of Obstetrics and Gynaecology, Kalafong Provincial Tertiary Hospital, Pretoria, South Africa, 3 Department of Radiation Oncology, University of Pretoria, Pretoria, South Africa, 4 Department of Radiation Oncology, Steve Biko Academic Hospital, Pretoria, South Africa, 5 Department of Obstetrics and Gynaecology, Gynaecologic Oncology Unit, Steve Biko Academic Hospital, Pretori, South Africa

* adekunlesajo@yahoo.com

## Abstract

### Objective

The main objective of this study was to describe the clinical, pathological and treatment characteristics of patients with vulvar cancer who had surgery and or radiotherapy at Steve Biko Academic Hospital. The absolute and relative disease burden, trends over the time period were also analyzed.

### Materials and methods

This was a retrospective study that described women with vulvar cancer who presented to the Gynaecology Oncology and Radiation Oncology departments of the hospital between January 2012 and December 2022. Their clinical, pathology and radiotherapy treatment records were reviewed for this study.

### Results

317 vulvar cancer cases between 2012 and 2022 were included in the analysis. The average age was 45.1±12.7. Forty percent of the participants were younger than 40 years. More than 75% of them were women living with HIV and were all on antiretroviral treatment. The average age of those who were HIV positive was 20 years lower than their HIV negative counterparts, p <0.0001. Their mean haemoglobin (Hb) at presentation was 10.7g/dL. Squamous cell carcinoma was the most common histological type in 96.5% of cases. Sixty four percent of the cases presented in advanced stage. About 48% of those who had primary radiation received curative doses. The median time to completion of radiotherapy treatment was higher among those who received primary radiotherapy as compared to those who received adjuvant

**Data availability statement:** All relevant data are within the paper and its Supporting information files.

**Funding:** The author(s) received no specific funding for this work.

**Competing interests:** The authors have declared that no competing interests exist.

treatment, 51.5 vs 46 days, p 0.039. The annual average age decreased from 56 years to as low as 40 years, a mean difference of 16 years, p 0.012.

## Conclusion

It is striking that vulvar cancer is no longer a disease of elderly women. Its incidence is now high among women below 50 years. The study also showed an upward trend in the number of vulvar cancer cases in contrast to the declining average age at diagnosis. There is need for more vulvo-perineal surveillance of HIV positive women to detect early stage of vulvar cancers.

---

## Introduction

Vulvar cancer though rare, accounts for 3–5% of all gynecologic malignancies. Globally, it accounted for over 45 000 new cases and 17 000 deaths in the 2020 global cancer estimate with age-standardized incidence rate (ASIR) and mortality of 0.9 and 0.3 respectively [1]. It shares common risk factors with cervical cancer such as advanced age, smoking, high risk sexual practices, HPV infection and immune suppression. Over 90% of vulvar cancers are squamous cell cancer subtypes, with a smaller percentage made up by the uncommon histological variants like sarcoma and clear cell cancers [2–6]. The predominant age group and ASIR differ between countries, with the Southern (1.5) and East Africa (1.3) having a higher ASIR for young women while Western Europe, North America and Australia demonstrate higher ASIR for women over 50 years [7–12]. The incidence of vulvar cancer has however been rising among premenopausal women albeit due to the high prevalence of HPV and co-infection of Human immunodeficiency virus (HIV).

In South Africa, the ASIR and age standardized mortality rate (ASMR) increased by 16.1% and 2.6% respectively between 1994 and 20213 [13]. The median age at diagnosis decreased by 18 years during same period. Butt and colleagues in South Africa also noted that their cohorts were 15 years younger than the age of women with vulvar cancer in developed world [9].

Surgery, ranging from simple wide local excision to radical vulvectomy with or without inguinofemoral lymphadenectomy, is the main treatment modality for early stage diseases. Radiotherapy and chemotherapy also play significant role in advanced disease and adjuvant treatment. Adjuvant therapy is often offered post-surgery to reduce the risk of recurrence in women with adverse pathological features. About 10–40% of vulvar cancers recur with majority of the recurrences occurring within 2 years of completing primary treatment [2,5,14,15].

Vulvar cancer has been largely understudied globally. Data on the demographics, pathologic distributions and outcomes of women diagnosed with vulva cancer in sub-Saharan Africa is sparse. Africa relatively has younger populations as compared to the western world. Therefore, we suspect a different demographics of women with vulvar cancer in African subregions. This study would provide some insight into the knowledge of the demographic, pathologic features and trends of vulvar cancer in South Africa.

## Materials and methods

This was a retrospective cross-sectional descriptive analysis of all cases of vulvar cancer between January 2012- December 2022 at the at the Gynaecologic Oncology Unit at Steve Biko Academic Hospital (SBAH). Data were accessed between 31st July 2023 and 3rd January 2024. Ethics approval (reference: 309/2023, dated 29th June 2023) was obtained from the University of Pretoria Faculty of Health Sciences Research Ethics Committee. All data were extracted from the medical records obtained from the departments of Gynecology, and Radiation Oncology. These included: age, parity, comorbidities disease stage and localization, HIV related details, pathological features and management. Laboratory data was obtained via the National Health Laboratory Services (NHLS Labtrak) electronic database. The 2009 FIGO staging was used to stage all patients for uniformity [16]. Surgery comprised complete/radical vulvectomy, wide local excision or partial/hemi-vulvectomy. The use of unilateral or bilateral inguinofemoral lymphadenectomy was dependent on the size, depth and laterality of the lesion. Primary chemoradiation was offered to those deemed inoperable while adjuvant radiotherapy was given to those who were at high risk for recurrence after surgery. The criteria for adjuvant treatment included: positive nodes, close or involved margins and others features as determined/set by the multidisciplinary tumour board. Stata 18 statistical package (StataCorp LP, Texas, USA) was used for data cleaning and analysis. The demographic and clinicopathologic characteristics of the participants were presented as frequencies, tables and graphs. Continuous variables were checked for normality using the skewness-Kurtosis test. Variables that were normally distributed were presented as mean ± standard deviation, while those that were not distributed normally were presented as median and interquartile range. Pearson's Chi-squared test (or Fischer's exact test when the expected cells were less than 5 in at least 25% of the cells) was used to determine the associations between binomial and categorical variables. The Student's t-test or Wilcoxon rank sum test was used to compare age, parity, Hb level, tumour size, and CD4 count across categorical variables with two levels. One-way Analysis of Variance (ANOVA) was used to compare the mean age across the years of diagnosis, followed by the Bonferroni post hoc test. The univariable and multivariable logistic regression analysis was used to assess the predictors of lymph node metastasis. Variables with p-value <0.2 were used in a stepwise regression fashion to build the final multivariable model. The Pearson's goodness of fit (Hosmer and Lemeshow's goodness of fit test) was conducted for the final multivariable model to determine the degree of fit of the data. A model to which the variables fit the data is expected to have a P-value >0.05. The AUB was also determined to assess the discriminatory value of the model. The level of statistical significance was set at p-value <0.05, confidence interval of 95% and the two–tailed test of hypothesis was assumed.

## Results

The data analysis included 317 participants with vulvar cancer treated between January 2012 and December 2022. From the 404 files that were evaluated, 87 were excluded due to incomplete or poor records and those who were erroneously coded as vulvar cancer.

### Demographic and clinical characteristics

The mean age of the participants was 45.1 ± 12.7 (median age 42 (IQR 37–51). The youngest patient was 14 years while the oldest was 87 years (Table 1).

Forty percent of the participants were younger than 40 years and over 75% of the participants were women living with HIV. Among those who were below 40yrs of age, over 95% of them were women living with HIV while those who were above 60yrs of age were those predominantly without HIV. Those living with HIV were statistically significantly younger than those who were HIV negative, p < 0.001. The proportion of those with haemoglobin (Hb) less than 10g/dL at presentation was significantly higher among those who were living with HIV as compared to the HIV negative counterparts, 87.6% vs 12.4%, p < 0.001. Eighty two percent of the participants did not have other medical comorbidity such as diabetes,

**Table 1. Demographic and clinical characteristics.**

| Variable | | HIV neg (69 = 21.8%) | HIV pos (248 = 78.2%) | p value |
|---|---|---|---|---|
| **Age** (years), **45.1±12.7** | | **60.6±14.7** | **40.8±8.0** | **<0.0001[a]** |
| Below 40 | | 3 (4.4) | 125 (50) | <0.001[a] |
| 40-49 | | 11 (15.9) | 89 (35.9) | |
| 50-59 | | 16 (23.2) | 28 (11.3) | |
| 60 and above | | 39 (56.5) | 6 (2.4) | |
| **Parity, 2(IQR:1–3)** | | **3 (IQR-2–3)** | **2 (IQR-1–3)** | **<0.0001*** |
| 0 | | 7 (12.5) | 27 (12.1) | <0.001[c] |
| 1-4 | | 35 (62.5) | 184 (82.5) | |
| ≥5 | | 14 (25) | 12 (5.4) | |
| **HB (g/dl), 10.7±2.4** | | **11.6±2.5** | **10.4±2.3** | **0.0009[a]** |
| <10 | | 13 (19.7) | 92 (38.5) | 0.001[c] |
| 10-11.9 | | 17 (25.8) | 78 (32.6) | |
| ≥12 | | 36 (54.5) | 69 (28.9) | |
| **Comorbidity** | | | | |
| DM | | 6 (8.8) | 4 (1.6) | <0.001[+] |
| HTN | | 19 (27.9) | 13 (5.26) | |
| Cardiac | | 2 (2.9) | 1 (0.4) | |
| CKD | | 2 (2.9) | 1 (0.4) | |
| DM & HTN | | 4 (5.9) | 3 (1.2) | |
| None | | 35 (51.5) | 226 (91.1) | |
| **Performance status (ECOG)** | | | | |
| 0-1 | | 48 (69.6) | 141 (56.9) | 0.057[c] |
| 2-4 | | 21 (30.4) | 107 (43.1) | |
| **Adjacent Organ involved** | | | | |
| Perianal | | 6 (60) | 32 (45.7) | 0.37[c] |
| Pelvic bone | | 0 (0.0) | 5 (7.1) | |
| Urethra | | 3 (30) | 12 (17.1) | |
| Vagina | | 1 (10) | 21 (30) | |
| **Tumour size (cm), 5.4±2.6** <br> **Range (0.5–18 cm)** | | n = 45, 4.7±1.5 | n = 122, 5.7±2.9 | 0.026[a] |
| <2 | | 1 (1.5) | 15 (6.2) | 0.004[c] |
| 2-4 | | 19 (27.5) | 30 (12.4) | |
| >4 | | 49 (71.0) | 198 (81.4) | |
| **Follow up (months), 18(IQR:6–48), 28.5±28** | | 23 (IQR:9–70) | 17 (IQR:6–38) | 0.036* |
| **FIGO stage** | Early | 32 (46.4) | 82 (33.1) | 0.042[c] |
| | Advanced | 37 (53.6) | 166 (66.9) | |
| **Histological types** | SCC (**96.5%**) | 62 (89.8) | 244 (98.4) | 0.006[+] |
| | Sarcoma | 4 (5.8) | 2 (0.8) | |
| | NEC | 1 (1.5) | 1 (0.4) | |
| | Adenocarcinoma | 1 (1.5) | 1 (0.4) | |
| | Paget's disease | 1 (1.5) | 0 (0) | |
| **Grade** | 1 | 6 (10) | 20 (8.9) | 0.46[c] |
| | 2 (**75.8%**) | 48 (80) | 168 (74.7) | |
| | 3 | 6 (10) | 37 (16.4) | |
| **Focality** | unifocal | 47 (68.1) | 126 (50.8) | 0.011[c] |
| | multifocal | 22 (31.9) | 122 (49.2) | |

*(Continued)*

**Table 1.** (Continued)

| Variable | | HIV neg (69=21.8%) | HIV pos (248=78.2%) | p value |
|---|---|---|---|---|
| **Treatment type** | Radiation, n=114 | 17 (27.4) | 97 (47.6) | 0.003[c] |
| | Surgery, n=152 | 45 (72.6) | 107 (52.4) | |
| **Interval between biopsy & first clinic visit (weeks), 10.8±9.1** | | **8.9±5.7** | **11.3±9.8** | **0.055** |
| ≤6 weeks | | 28 (42.4) | 77 (32.1) | 0.12[c] |
| ≥6 weeks | | 38 (57.6) | 163 (67.9) | |
| **CD4 count (cell/µL), 481.8±260** | <200 | | 28 (12.2) | |
| | 201-499 | | 100 (43.9) | |
| | ±500 | | 100 (43.9) | |
| **Viral load (copies/mL** | Lower than detectable | | 99 (60.4) | |
| | Suppressed (<1000) | | 36 (21.9) | |
| | Unsuppressed (≥1000) | | 29 (17.7) | |
| **Interval between ARV initiation and cancer diagnosis** | | | | |
| <10yrs | | | 90 (71.4) | |
| ≥10yrs | | | 36 (28.6) | |

[a]Student t test, [c]Chi-squared, [*]Mann-Whitney U test, [+]Fishers Exact.

hypertension or cardiorenal diseases. Those who were HIV positive had significantly larger tumour size as compared to those without HIV, 5.7 cm vs 4.7 cm, p 0.026. Sixty four percent of the cases presented in advanced stage (stage III and IV). Among those who presented in late stage, 81% of them were women living with HIV. About 55% of the disease were unifocal. A greater proportion (57%) of the patients had surgery as the first modality of treatment. Among those who were HIV positive, the mean CD4 count was 481.8±260 cell/µL while 82.3% of them either had lower than detectable or suppressed viral load.

## Pathological characteristics of patients who had surgery

The histopathologic features of those who had surgery are shown in Table 2.

About two-thirds (62.5%) of the tumours were larger than 4 cm in the greatest dimensions. Among those with sarcomas, 2 of them were younger than 18 years with histological sub-type of Rhabdomyosarcoma. Only about a fourth of the disease involved more than one site. Among those who had nodal dissection, over 95% of them had bilateral inguinofemoral nodal dissection. Thirty five percent of the surgical free margins were 8 mm or more while vulvar intraepithelial neoplasia (HSIL) was present at the surgical margin in about half of the surgical specimens.

## Treatment modalities

Of the 152 patients who had surgery, about 70% had radical vulvectomy, Table 3.

Of the 165 patients who were referred for primary radiotherapy, about 70% of them received treatment with either curative or palliative intent. Those who were not treated were either too sick to receive treatment or failed to present to the radiotherapy department. Forty seven percent of those who were treated received curative/radical doses of radiation, with or without chemotherapy while 52.6% of them received palliative radiation due to their advanced incurable disease. Those who did not receive adjuvant treatment either refused to go after surgery despite referral or presented at a later time when adjuvant radiotherapy was no longer considered beneficial. The average time interval between referral from gynaecologic oncology clinic and commencement of radiotherapy treatment was 18.9±11 weeks. The median time to completion of radiotherapy treatment was 48.5 days (IQR: 35.5–64).

**Table 2. Histopathological characteristics of surgical specimens.**

| Features | | N = 152(%) |
|---|---|---|
| **Tumour size** | | 5.0 ± 1.9 |
| ≤4 cm | | 57(37.5) |
| 4.1-6.0 cm | | 69 (45.4) |
| 6.1-8.0 cm | | 20 (13.1) |
| 8.1-10 cm | | 3 (2.0) |
| >10 cm | | 3 (2.0) |
| **focality** | Unifocal | 111 (73.0) |
| | Multifocal | 41 (27.0) |
| **Histologic type** | | |
| SCC | | 147 (96.7) |
| Sarcomas | | 3 (1.9) |
| Adenocarcinoma | | 1 (1.0) |
| Paget's disease | | 1 (1.0) |
| **Grade** | 1 | 15 (10.9) |
| | 2 | 107 (77.5) |
| | 3 | 16 (11.6) |
| **Lymph node dissection** | | |
| ipsilateral | | 3 (2.0) |
| Bilateral | | 120 (81.2) |
| Not performed | | 29 (16.8) |
| **Number removed** | | 14.9 ± 6.5, range: 2–36 |
| <12 | | 40(32.5) |
| ≥12 | | 83(67.5) |
| **Nodal status** | Negative | 79 (64.2) |
| | Positive | 44 (35.8) |
| **Number of positive nodes** | | 2 (IQR 1–3) |
| 1 | | 15 (34.1) |
| 2 | | 8 (18.2) |
| ≥3 | | 21(47.7) |
| **Extracapsular spread** | No | 29 (65.9) |
| | Yes | 15 (34.1) |
| **Surgical margin (mm)** | | |
| <3 | | 31 (20.4) |
| 3-4 | | 38 (25.0) |
| 5-7 | | 29 (19.1) |
| ≥8 | | 54 (35.5) |
| **Depth of infiltration (mm)** | | |
| ≤1 mm | | 17 (15.7) |
| 2-3 mm | | 23 (21.3) |
| >3mm | | 68 (63.0) |
| **VIN present at margin** | No | 77 (51.7 |
| | Yes | 72 (48.3) |
| **LVS** | No | 47 (30.9) |
| | Yes | 10 (6.6) |
| | Not reported | 95 (62.5) |

*(Continued)*

**Table 2.** (Continued)

| Features | | N = 152(%) |
|---|---|---|
| Perineural involvement | No | 26 (17.1) |
| | Yes | 4 (2.6) |
| | Not reported | 122 (80.3 |

**Table 3. Characteristics of treatment modality.**

| Primary treatment received | | | |
|---|---|---|---|
| **Surgery** | | **N = 152, (%)** | |
| Partial vulvectomy | | **32 (21.0)** | |
| Radical vulvectomy | | **105 (69.1)** | |
| Wide local excision | | **15 (9.9)** | |
| **Interval between first visit and surgery** (weeks) | | **4 (IQR:3–11)** | |
| **Radiotherapy (Primary)** | | **N = 165, (%)** | |
| | Treated | **114 (69.1)** | |
| | Not treated | **51 (30.9)** | |
| **Treated** | Curative/radical | **54 (47.4)** | |
| | Palliative | **60 (52.6)** | |
| **Adjuvant RT** | Given | **29 (42.7)** | |
| | Not given | **39 (57.3)** | |
| **Interval between Referral and start of RT** (weeks) | | | |
| **18.9 ± 11**, 20 (IQR:9–29) | **primary** | **Adjuvant** | **P value** |
| | 20 (IQR:9–30) | 16 (7-20) | 0.017* |
| **Radiotherapy completion time (days)** | | | |
| | **Standard ≤56 days** | **Prolonged >56 days** | **P value** |
| 48.5 (IQR:35.5–64) | 51.5 (IQR:40–70) | 46 (IQR 25–56) | 0.039* |
| **primary** | 31 (57.4) | 23 (42.6) | 0.18c |
| **Adjuvant** | 21 (72.4) | 8 (27.6) | |

cChi-squared, *Mann-Whitney U test.

## Trends of vulvar cancer over the study period

Fig 1 showed that the numbers of vulvar cancer per year increased from 10 in 2012–33 in 2022 with some marked upward variations.

There was a significant variation in the number of cases across the years of study. The average age per year decreased from 56 years to as low as 40 years with a mean difference of 16 years, p 0.012. After Bonferroni post-hoc test, the difference lied between 2012 and 2018 years of diagnosis. The mean age of the patients showed downward trend while the number of cases also showed increasing upward trend with some marked variations between the period 2015 and 2017. When the years of diagnosis were halved, the mean age at diagnosis in the first half (2012–2016) was significantly higher than that of the latter half, 46 years vs 44 years, p 0.001.

As shown in Fig 2a, b, the numbers of women with vulvar cancer who were living with HIV showed significant upward trend across the study years, p 0.001.

The percentage distribution of HIV positivity among the women increased from 20% in 2012 to as high as 96% in 2021.

A

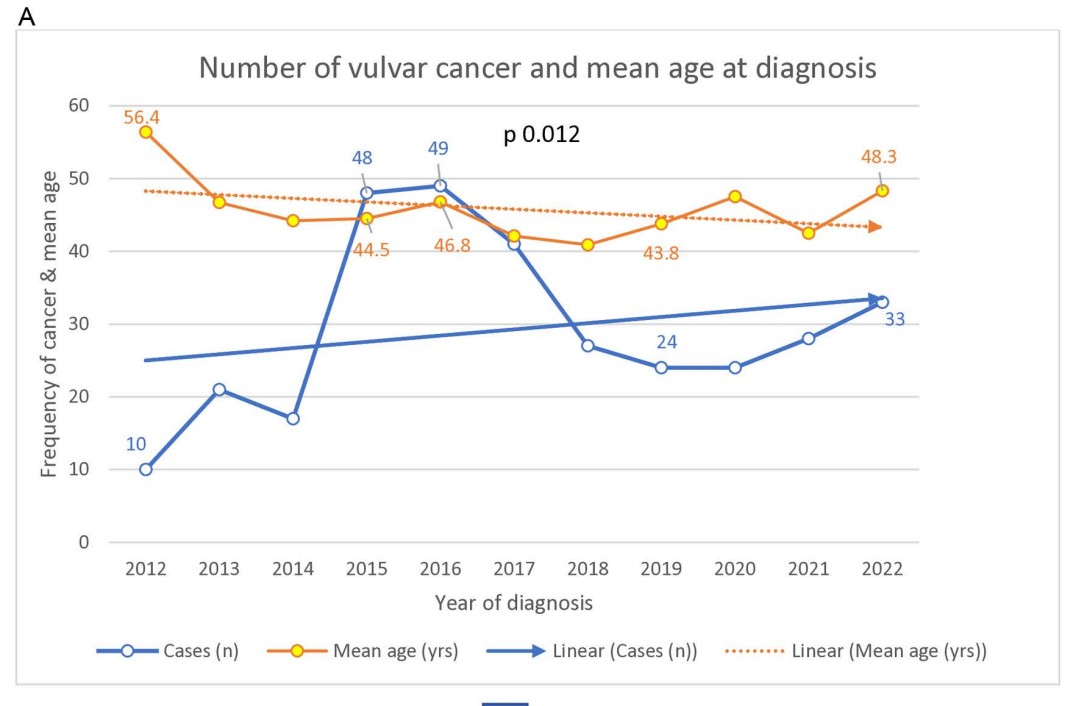

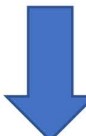

B

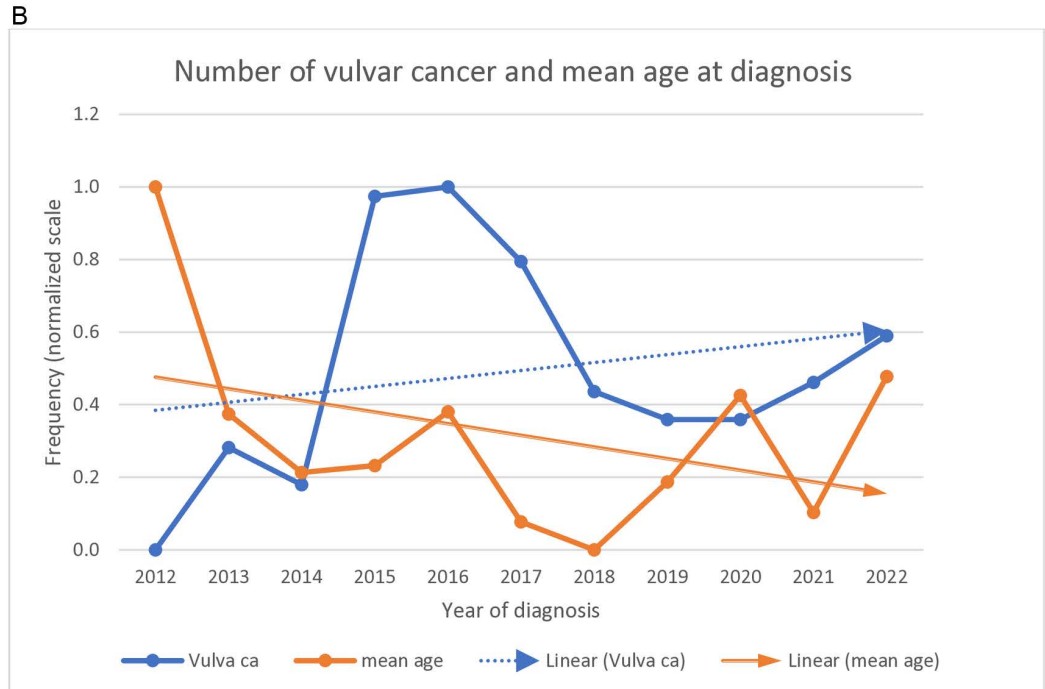

**Fig 1. (a,b) Trend of vulvar cancer and mean age at diagnosis.**

A

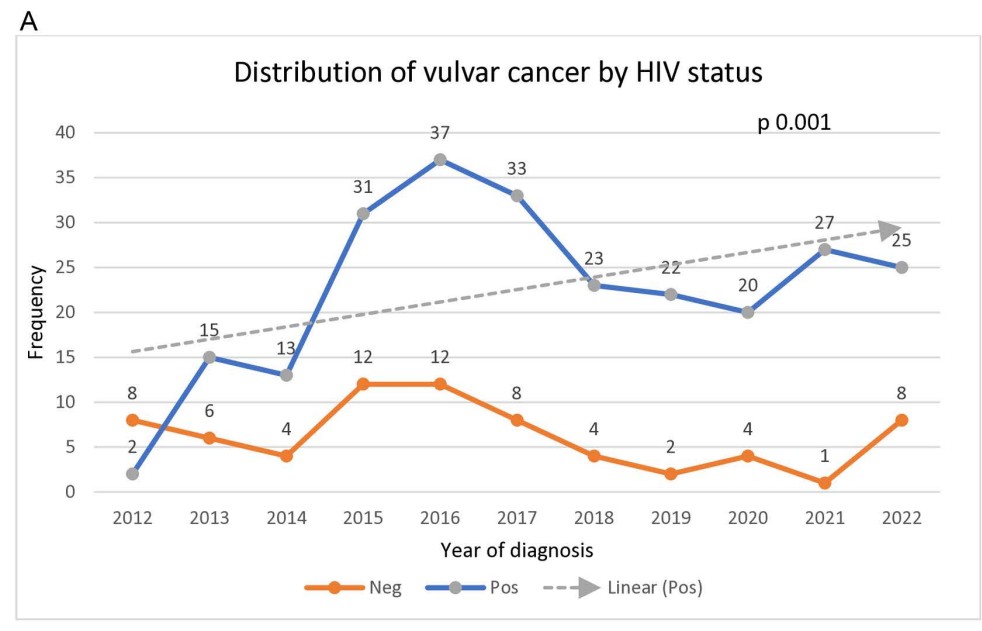

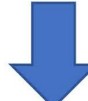

B

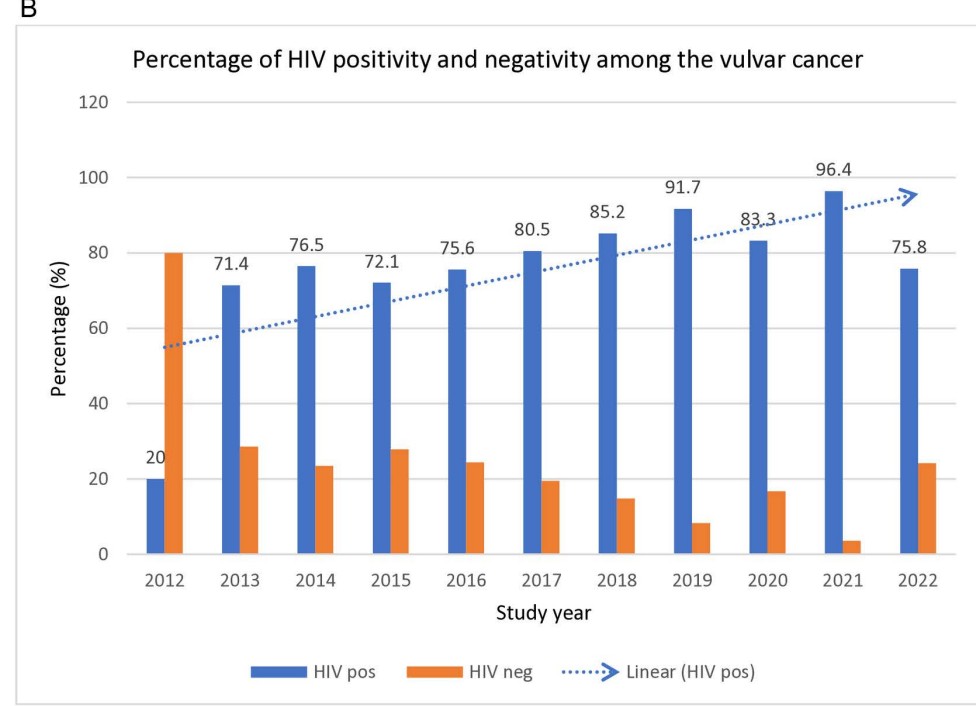

**Fig 2. (a,b) Distribution of vulvar cancer by HIV status.**

There was no difference in the stage of disease at presentation across the study years, p 0.45.

During the study year, the number of cervical cancers showed downward trends relative to the upward trends of the number of vulvar cancers, Fig 3a, b.

Between 2016 and 2019, both cancers followed the same pattern of distribution per year, Fig 3b.

## Predictors of nodal metastasis among surgical group

From Table 4, the univariate logistic regression analysis showed lymphovascular space involvement and haemoglobin less than 10g/dl were individually predictive of lymph node metastasis among those who had surgical treatment.

Lymphovascular space involvement had seven folds increased odds of predicting lymph node metastasis in univariate analysis. Tumour size of greater than 4 cm and poor tumour grade showed trends towards significance in predicting lymph node metastasis, p 0.06 vs p 0.05 respectively. After adjusting for confounding variables such as tumour size, HIV status, lymphovascular space involvement and depth of invasion, poor tumour grade and haemoglobin levels of less than 10g/dl had 5 folds odds of predicting lymph node metastasis (a marker of advanced disease), p 0.049 vs 0.036 respectively. The Hosmer-Lemeshow test showed that the post estimation p-value was 0.07. Since this p-value was greater than 0.05, this means that our model fits the data reasonably well. The area under the curve of the receiver operating characteristic was 0.77, which suggested that the model was fairly predictive of inguinofemoral nodal metastasis.

## Discussion

Our study found that the mean age of vulvar cancer was 45 years and a greater proportion of the women were younger than 50 years. While the mean age at diagnosis has been decreasing, the number of vulvar cancers per year has been rising significantly in the past decade of the study. More than three-quarters of the patients were women living with human immunodeficiency virus (HIV), one of the highest ever documented from a single institution in the literature. Those who were HIV positive were, on the average, 20 years younger than the HIV negative population. Squamous cell carcinoma was the most common histological type which is similar in proportion to what had been described in other studies.

In southern Africa, vulvar cancer is fast becoming a non-rare disease entity in view of the rising incidence especially among younger women. Even though the age at diagnosis has been reducing globally, the average age of women with vulvar cancers in our population has significantly moved downwards away from the 7th and 8th decades benchmark quoted in many literature as the age at diagnosis [5,17–20]. The mean age of 65 years reported in a Tunisian study was closer to the average age of above 60 years that was reported in most high-income countries that are predominantly of white and miscegenetic descents [21]. But similar studies by Loggenberg and Butt et al in the Western Cape of South Africa, which is predominantly of white and coloured races, reported average ages of 52 and 54 years respectively [8,9]. In Asia, Europe and the USA, the average age at diagnosis ranged between 65 and 72 years [18,22–25]. Most of the studies from the countries did not report the proportions of HIV in their cohort which reflects the low prevalence of HIV that would have allowed for comparison. However, studies from sub-Saharan Africa, especially the Southern African countries with high prevalence of HIV, reported the proportions of HIV between 20% and 89% in their study cohorts [8,9,25–28]. The reported mean age of 45 years in this study is similar to the average ages of 42 and 45 years reported in studies from Botswana and Mozambique which are neighbouring countries to South Africa to the North West and North East respectively [25,27]. The average age of vulvar cancer appears to have stabilized between 40 and 45 years in our institution. A 12-year review of cases of advanced vulvar cancer by Mnisi et al reported a median age of 43 years [29]. Aside the methodological differences between our study and Mnisi et al that reported on data between 2001 and 2013 from two academic hospitals, the two years overlap of the study period did not affect the age comparison. It is striking that 40% of our study population were younger than 40 years. Chikandiwa et al reported an increase of 1.7 per 100 000 in age specific incidence rate of vulvar cancer among women between 30–39 years in the 2009–2012 period as compared to an increase

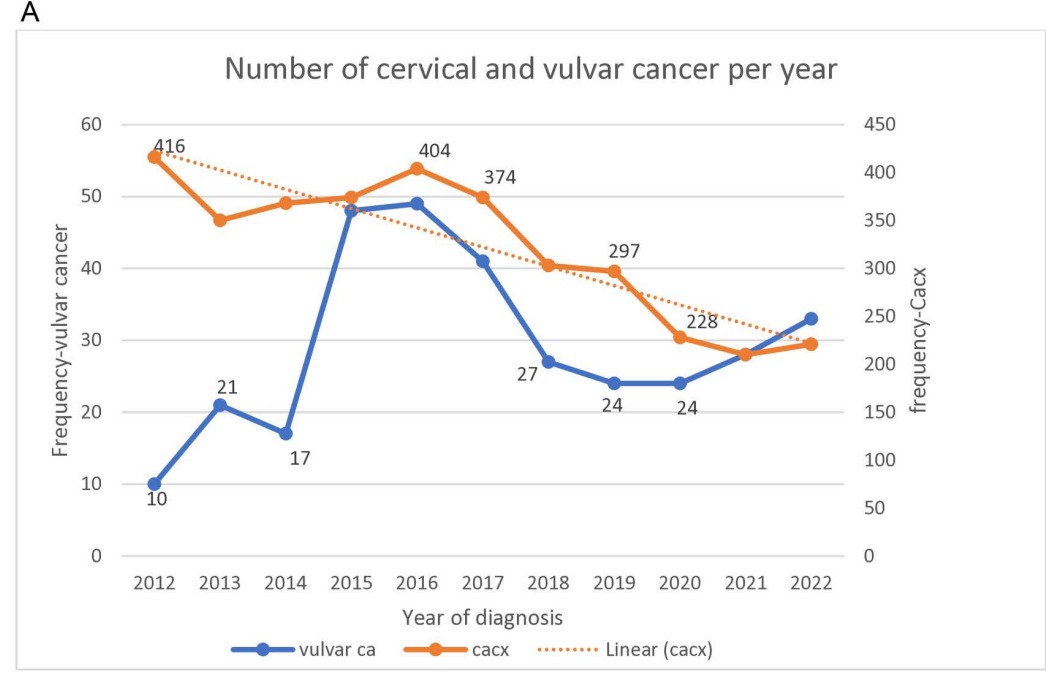

A

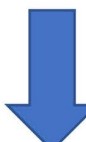

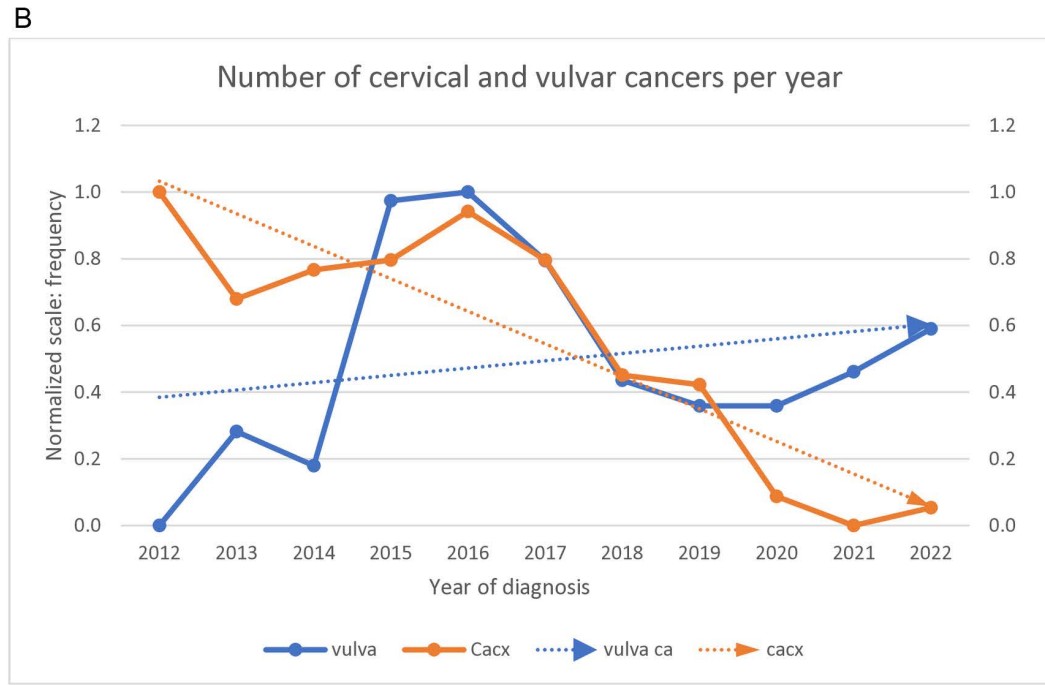

B

**Fig 3. (a,b) Distribution of cervical and vulvar cancers during the period of study.**

**Table 4. Univariate and Multivariate Logistic Regression for the predictors of nodal metastasis among the operated patients.**

| Variables | OR | 95% CI | p value | Adj OR | 95% CI | p value |
|---|---|---|---|---|---|---|
| **Age** | | | | | | |
| 50yrs and above | 1.00 | reference | reference | 1.00 | reference | reference |
| Below 50yrs | 1.17 | 0.54-2.58 | 0.69 | | | |
| **Tumour size (cm)** | Coef. 0.27 | 0.05-0.48 | 0.015* | | | |
| <4 | 1.00 | reference | reference | 1.00 | reference | reference |
| >4 | 2.11 | 0.95-4.70 | 0.06 | 2.9 | 0.73-11.3 | 0.13 |
| **HIV status** | | | | | | |
| Negative | 1.00 | reference | reference | 1.00 | reference | reference |
| Positive | 1.67 | 0.71-3.91 | 0.24 | 1.15 | 0.33-3.97 | 0.83 |
| **Focality** | | | | | | |
| Unifocal | 1.00 | reference | reference | 1.00 | reference | reference |
| multifocal | 1.47 | 0.65-3.34 | 0.35 | – | – | – |
| **Duration of ARV use** | | | | | | |
| <10yrs | 1.00 | reference | reference | 1.00 | reference | reference |
| 10yrs or more | 1.81 | 0.47-6.97 | 0.39 | – | – | – |
| **CD4 count (cells/µL)** | | | | | | |
| <200 | 1.00 | reference | reference | 1.00 | reference | reference |
| 200-499 | 0.57 | 0.12-2.71 | 0.48 | – | – | – |
| ≥500 | 0.64 | 0.14-2.93 | 0.57 | – | – | – |
| **Viral load** | | | | | | |
| Lower than detectable | 1.00 | reference | reference | 1.00 | reference | reference |
| Suppressed | 2.7 | 0.70-10.65 | 0.15 | – | – | – |
| Unsuppressed (≥1000) | 0.39 | 0.07-2.04 | 0.26 | – | – | – |
| Haemoglobin (g/dl) | | | | | | |
| ≥12 | 1.00 | reference | reference | 1.00 | reference | reference |
| 10-11.9 | 1.87 | 0.78-4.50 | 0.16 | 2.26 | 0.57-8.95 | 0.24 |
| <10 | 4.7 | 1.6-13.4 | 0.004* | 5.0 | 1.1-18.4 | 0.036* |
| **Lymphovascular space** | | | | | | |
| No | 1.00 | reference | reference | 1.00 | reference | reference |
| Yes | 7.0 | 1.4-35.4 | 0.019* | 1.85 | 0.24-14.2 | 0.55 |
| **Perineural involvement** | | | | | | |
| No | 1.00 | reference | reference | 1.00 | reference | reference |
| Yes | 1.87 | −0.59 −4.33 | 0.14 | – | – | – |
| **Tumour Grade** | | | | | | |
| Good | 1.00 | reference | reference | 1.00 | reference | reference |
| Poor | 3.05 | 0.99-9.33 | 0.05 | 5.1 | 1.00-25.6 | 0.049* |
| **Depth of Infiltration (mm)** | | | | | | |
| ≤1 mm | 1.00 | reference | reference | 1.00 | reference | reference |
| 2-3 mm | 1.71 | 0.15-19.4 | 0.66 | 1.46 | 0.09-22.6 | 0.79 |
| >3mm | 5.9 | 0.69-50.1 | 0.11 | 4.94 | 0.45-54.6 | 0.19 |

OR= Odd ratio, Adj. OR = adjusted odd ratio, CI: 95% Confidence interval, *significance.

of 0.1 in the period 1994–1998 [13]. In contrast to findings from the Southern Africa region, studies from West African countries reported average age at vulvar cancer diagnosis ranging between 48 and 61 years [30,31].

Human immunodeficiency virus is prevalent in most sub-Saharan African countries. South Africa is one of the countries in the world with the highest proportion of women living with HIV. The etiological factors associated with vulvar cancer is somewhat geographical. It is reported that most vulvar cancers in Africa are associated with HPV as against the HPV-independent vulvar cancers in high income countries [8,9,13,25]. HPV and HIV play synergistic role in the dysplastic changes within the epithelial lining of the lower anogenital tract. The rising incidence of vulvar cancer among younger women is therefore related to the increase in HIV and HPV coinfections in South Africa [8,9,13,25]. Over 75% of the women in our study were HIV positive. This is one of the highest proportions ever documented in the literature. It is also higher than the prevalence of 40% reported in our institution a decade earlier than the period covered in our study [29]. In Botswana studies, the proportions of HIV positivity among the cohorts of women with vulvar cancer ranged between 56.9 and 89% [27,28]. The 25% national prevalence of HIV among Batswana women in the reproductive age in these studies was similar to the statistics from South Africa [32]. Aside from the population difference, both countries share similar demographics, hence, it is not surprising that the mean age of 40 years among the HIV positive women in our study is similar to the median age of 41 years reported in Botswana among those living with HIV who had vulvar cancer. In our cohorts, the proportion of HIV in those younger than 50 years was 93.8%. All the WLWH in the present study were on ARVs with a median interval between the commencement of ARV and vulvar cancer diagnosis of 6 years. This was relatively similar to the interval of 9 years reported by MacDuffie and coworkers. Most of the patients were relatively immunocompetent, mean CD4+ of 481 cells/µL as well as being virally suppressed. Hence, being immunocompetent and virally suppressed from chronic use of ARV might not positively influence the initiation and progression of HPV related vulvar dysplasia or there might be another pathway for the carcinogenesis of vulvar cancer in WLWH.

Furthermore, apart from the influence of HIV on the age at diagnosis of vulvar cancer, it also impacts on the clinicopathologic characteristics of women with vulvar cancer. Not only was the mean haemoglobin level lower, the tumour size was significantly larger in WLWH as compared to their counterparts. The lower haemoglobin could be due to persistent bleeding from larger tumours, chronic ARV use and from the effect of HIV on haemopoiesis. MacDuffie et al did not find any impact of HIV on the haemoglobin levels of patients with vulvar cancer. Most other literature did not describe the impacts of HIV on the clinicopathologic features of vulvar cancer. Hence, there is a need for a prospective study in our institution to describe this impact.

The most prevalent histological type in our study was squamous cells carcinoma (SCC) which accounted for over 95%. This agrees with the findings in most literature where SCC was reported as the commonest histological type [2,21]. Vulvar squamous cell carcinoma is dichotomous. It can either be HPV dependent or HPV non-dependent, each arising from two different premalignant lesions. The most common vulvar SCC, basaloid or warty type, is mostly driven by high risk HPV infection and has been found to have better prognosis [6]. Tumour size plays a significant role in vulvar cancer management and risk stratification. Surgical modality for larger tumours is associated with significant morbidity, compromise of vital contiguous structures, use of flap to close defects and risk of inadequate free margins. About eighty percent (79.2%) of our study population presented with tumour larger than 4 cm. In a Northern Africa study, the average tumour size was less than 4 cm [21]. The left vulva was the most affected site while the perianal area was the most involved contiguous site. This finding was different from the report by de Melo Maia et al where the vagina was the most affected adjacent structure in 15% of cases [17]. Unlike in America and other HICs, patients often present in advanced stage disease in Africa and other low and middle income countries (LMIC). Two-thirds of the patients seen during the study period presented in either stage III or IV. This often reflects poor access to healthcare, poor referral system, poverty, misdiagnosis and insufficient health education and awareness on the parts of the patients. Oncological services in South Africa, far better than in most other African countries, are regionalized in cities. Hence, patients often need to be transported from remote areas to access these services usually at an advanced stage. A reflection of these challenges was noticed in

our study where the average time between biopsy at local clinics and presentation at the regional specialty clinic was 10 weeks while the average time to commencement of radiotherapy was 18 weeks from the time of referral from the gynaecology specialty clinic.

Surgery, with variable modalities, is the standard form of treatment for vulvar cancer. Radiotherapy with or without chemotherapy is offered as the primary treatment for advanced disease and as adjuvant treatment for surgically treated patients who are at increased risk of disease recurrence. The general recommendation for a tumour not more than 2 cm is wide local excision with or without inguinofemoral lymphadenectomy depending on the depth of stromal invasion [2,15,17,21]. Of the 266 patients who were treated, 57% had surgery. This was in contrast to the study from Brazil where over 85% of their patients had surgery which might be related to the early stage presentation reported in majority of their cohorts [17]. Radical vulvectomy was the most frequently performed surgical modality in our study because the greater proportion of the surgical group presented with larger tumours. In Kehila et al, all their patients were surgically managed as none of them presented with stage IV disease [21]. About 70% and 42% of our study population were able to receive primary and adjuvant radiotherapy treatment respectively. The identifiable factors responsible for these proportions include prolonged waiting time to access radiotherapy services, poor clinical state, prolonged interval after primary surgery, refusal to show for treatment and loss to follow up. There is paucity of radiotherapy services across Africa which often results in an increased burden of referrals on the limited services. The median time to completion of primary radiotherapy for locally advanced vulvar cancer in our study was 52 days. The completion of radiotherapy within the scheduled time interval is a determinant that is associated with improvement in oncological survivals. However, this has only been demonstrated in locally advanced cervical cancer [24,33]. In a study by Song et al, a completion time beyond 56 days was associated with increased risk of pelvic recurrence in cervical cancer [33]. Although similar findings have not been well demonstrated in vulvar cancer despite the similarities between the two cancers in terms of carcinogenesis, behavior and treatments.

Lymph node metastasis is the most important independent negative prognostic predictors in vulvar cancer. There are several potential predictors of lymph node metastasis which define the stage and treatment of the disease. These include tumour size, poor grade, aggressive histological types, performance status, tumour localization, deep stromal invasion, lymphovascular space and perineural invasion [2,18,34,35]. Only poor tumour differentiation was predictive of lymph node metastasis in our study after controlling for confounders. Reports from other studies have been inconsistent in the factors defining nodal disease prediction [36–38]. In a Danish study of 388 vulvar SCC, increasing tumour size, tumour close to the clitoris and poor grade were all significantly associated with increased risk of inguinofemoral nodal metastasis [34]. The 35% lymph node positivity in our surgical cohorts was consistent with the findings of 28–38% nodal positivity in many other literature [18,34,39]. However, this was higher than the prevalence of 16% of positive nodes among 15 000 vulvar SCC from the USA National Cancer Database [37]. The low proportion of nodal positivity in this USA study might be related to the higher proportion of early stage disease among their cohorts.

Globally in the past few decades, vulvar cancers have been showing upward trends. Likewise, the average age at the onset of diagnosis has also been reducing. In our study, the number of vulvar cancers increased from 10 in 2012–33 in 2022 with significant variations between these years. Relative to 2012, our findings showed an upward trend in the number of cases across the years of study while the average age at diagnosis showed a linear downward trend. This trend was pronounced between 2012 and 2018 during which the average age decreased by 16 years. This finding is similar to a decline of 18 years reported over 8 years period by Chikandiwa and coworkers [13]. The incidence of vulvar cancer declined as the age of women increased beyond 50 years. This suggests an increase in HPV related vulvar cancers rather than the HPV independent types seen in older women. However, the decline in trend of cervical cancer during the study period might be due to the increase awareness and uptake of cervical cancer screening especially among WLWH. While Schuurman et al found a change in FIGO stage distribution especially with increase in stages I and III diseases over their study years, our finding did not show a trend away from the usual late disease stage at diagnosis [12]. The poor

help-seeking behavior, delay in referral by health professionals and prolonged appointments might be contributory to the lack of trend in disease stage at presentation. Also, the percentage positivity of HIV among the women increased from 20% in 2012 to the highest proportion of 96% in 2021. In recent times, we can almost predict with some certainty that a woman below 50 years who present with vulvar lesions suspected to be VIN or cancer, is living with HIV or other form of immunosuppressive condition.

The limitation of this study lies in the inherent drawbacks of retrospective study. Not all variables were complete for analysis. The impact of this on our findings might be limited as missing values were corrected for in the analysis. Also, the multi-file system within the institution might be the reason why some patients were reported as no show because the radiotherapy department uses different hospital numbers to code patients in their filings system. Hence, it is possible some patients were coded with different names and hospitals numbers different from the ones used during referral at the Gynaecologic Oncology Unit. But efforts were made to search the Radiotherapy Unit database using all available patients' names, hospital numbers and national identity number to reduce this limitation. The reasons why some of the known risk factors for nodal metastasis such as lymphovascular space, perineural invasion and depth of in invasion, did not reach significant levels in our study could be that some of these parameters were not uniformly reported by the pathologists.

However, the strength of our study lies in the large study population from a single academic hospital where the treatment patterns were similar. The pathology reports were centralized to the national laboratory database, making accessibility to be easy. Also, our study included 248 women that were living with HIV, the highest reported cohorts from a single institution in the literature. This provided us with insight into this special group of high-risk individuals in our population and the need to protect them.

In conclusion, it is striking that vulvar cancer is no longer a disease attributable to older aged women. Its incidence is now high among women below 50 years. This study has shown some insights into why women younger than 50 years could be susceptible to vulvar cancer. Despite being immunologically competent, women living with HIV are still at an increased risk of vulvar cancer or other malignancies. Hence, the need for more proactive measure to screen them not just for cervical cancer but also for vulvar cancer. The study will also provide information for health system planners about the current burden of the disease and how to mitigate its impacts. We recommend that women living with HIV should have at least an annual vulvo-perineal examination in addition to the regular health checks since they visit health care centers to collect their ARV medications. Public enlightenment on the signs and symptoms of early stage vulvar cancer would help in early detection and treatment to reduce its incidence and improve survival. The need for synchronized electronic medical records across the departments that run oncology services will help prevent missing data thereby improving clinical care and follow up. Our study is therefore an important additional to the growing body of evidence on vulva cancer in Africa.

## Supporting information

**S1 File. PLOSOne_Human_Subjects_Research_Checklist.**
(DOCX)

## Acknowledgments

We acknowledged Steve Biko Academic Hospital Medical Health Records, Radio-oncology and Gynaecologic clinic staff who assisted in retrieving the patients' medical records and my colleagues for their supports and encouragement.

## Author contributions

**Conceptualization:** Adekunle Emmanuel Sajo, Adekunle Emmanuel Sajo, Edwin Francis Mnisi, Sheynaz Bassa, Cathy Visser, Greta Dreyer.

**Data curation:** Adekunle Emmanuel Sajo, Adekunle Emmanuel Sajo, Sheynaz Bassa, Cathy Visser, Greta Dreyer.

**Formal analysis:** Adekunle Emmanuel Sajo, Adekunle Emmanuel Sajo.

**Investigation:** Adekunle Emmanuel Sajo, Adekunle Emmanuel Sajo.

**Methodology:** Adekunle Emmanuel Sajo, Adekunle Emmanuel Sajo, Edwin Francis Mnisi, Greta Dreyer.

**Project administration:** Adekunle Emmanuel Sajo, Adekunle Emmanuel Sajo.

**Resources:** Adekunle Emmanuel Sajo, Adekunle Emmanuel Sajo.

**Supervision:** Sheynaz Bassa, Greta Dreyer.

**Validation:** Adekunle Emmanuel Sajo, Adekunle Emmanuel Sajo, Edwin Francis Mnisi, Sheynaz Bassa, Greta Dreyer.

**Visualization:** Adekunle Emmanuel Sajo, Adekunle Emmanuel Sajo.

**Writing – original draft:** Adekunle Emmanuel Sajo, Adekunle Emmanuel Sajo, Greta Dreyer.

**Writing – review & editing:** Adekunle Emmanuel Sajo, Adekunle Emmanuel Sajo, Edwin Francis Mnisi, Sheynaz Bassa, Cathy Visser, Greta Dreyer.

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
