## [Decision Letter · Decision Letter 0]

Dear Dr. Sajo,

Thank you for submitting your manuscript to PLOS ONE. After careful consideration, we feel that it has merit but does not fully meet PLOS ONE’s publication criteria as it currently stands. Therefore, we invite you to submit a revised version of the manuscript that addresses the points raised during the review process.

We look forward to receiving your revised manuscript.

Kind regards,

Milad Khorasani, PhD

Academic Editor

PLOS ONE

Journal Requirements:

2. Please include captions for your Supporting Information files at the end of your manuscript, and update any in-text citations to match accordingly. Please see our Supporting Information guidelines for more information: http://journals.plos.org/plosone/s/supporting-information .

Reviewers' comments:

Reviewer's Responses to Questions

**Comments to the Author**

1. Is the manuscript technically sound, and do the data support the conclusions?

Reviewer #1: Yes

Reviewer #2: Partly

Reviewer #3: Yes

2. Has the statistical analysis been performed appropriately and rigorously?

Reviewer #1: Yes

Reviewer #2: No

Reviewer #3: Yes

3. Have the authors made all data underlying the findings in their manuscript fully available?

Reviewer #1: Yes

Reviewer #2: Yes

Reviewer #3: Yes

4. Is the manuscript presented in an intelligible fashion and written in standard English?

Reviewer #1: Yes

Reviewer #2: Yes

Reviewer #3: Yes

Reviewer #1: PlosOne Review NR 20/1/25

Clinico-pathological and treatment characteristics of HIV and non-HIV related vulvar cancers: Analysis of a South African cohort

Good study, with comprehensive data. Large HIV positive cohort adds value

Minor

1. Include the core diagnosis in the Abstract, Objective and Materials and Methods ie. vulvar cancer

2. Line 45: “Globally, it accounted for over 45 000 new cases and 17 000 deaths” What is the time period referred to?

3. Correct minor grammatical errors throughout: space between age and year, data was…, p value lower case in Table 1, spelling

4. Table 1: Is the mean or median tumour size documented? Is the p value for co-morbidity as a group or for DM only?

5. In the Discussion on Age, what % of patients in these referenced studies were HIV pos ie. were similar cohorts compared?

Major:

1. Stats in Table 1

Review stats in Table 1eg. HIV neg (69=21.8%) and HIV pos (248=78.2%) for the FIGO stage Early 32 (28.1) vs 82 (71.9) and Advanced 37 (18.3) vs 166 (81.8), with changing the denominator: Early HIV neg 32/69=46.4% vs 82/248=33% in HIV pos and Advanced HIV neg 37/69=53.6% and HIV pos 166/248=66.9%. The change in denominator allows for comparison of HIV pos and HIV neg groups, as the large number of HIV pos cases is a study strength.

2. Remove replication of Results described in words, when already presented in Tables

Reviewer #2: The study investigated the clinical, pathological, and treatment characteristics of patients of HIV and non-HIV-related vulvar cancers. However, my concerns/suggestions are not necessarily a reflection of the quality of the work as the following:

• Although this may seem like a minor issue, the abstract lacks depth and feels somewhat diluted. It merely touches on the topic without offering substantial insights. The manuscript is submitted as a research article, yet its primary objective aligns with a review perspective. Please refine the abstract and introduction to make it more concise, informative, and aligned with the intended research focus.

• Some comments about figures/tables here and there!

o In Figure 1:

The chart could benefit from data labels for individual points, especially for key inflection years like 2016 or 2019.

The notation is somewhat ambiguous; clarifying which trend it applies to would improve interpretability.

Using exponential for cases and linear for mean age may confuse readers unless justified. A brief explanation of why these particular trends were chosen would add clarity.

The chart lacks contextual information such as the type of cases being studied, geographical scope, or contributing factors that might explain the trends.

Consider normalizing the scales for easier visual comparison of the trends.

o In Figure 2:

The chart does not provide information about the population size or geographic scope. Are these proportions relative to the general population, specific cohorts, or testing populations?

The y-axis labeled “Proportion of HIV status” is vague. Are the proportions normalized to a specific population size or expressed as percentages?

The chart does not stratify cases by other variables such as age, gender, or risk factors, which could provide deeper insights.

o In figure 3:

The chart does not specify whether the frequencies are raw counts or adjusted for population growth, age standardization, or other factors.

There is no indication of geographic or demographic focus.

The dual y-axes might confuse some viewers, as it is not immediately clear that each cancer is plotted on a different scale.

o Their uploaded file, named as Excel sheet for PLoS ONE manuscript Nov 5, 2024, has some areas of discrepancy such as:

The Excel sheet includes several derived variables (e.g., tsize_cat5, age_cat9), which may need further clarification to match the previously provided raw data tables, reflecting the information's depth!

Some columns, like trend and multiple categorizations (e.g., agecat3, age_cat8, age_cat9), may not directly align with the manuscript’s core tables and could require filtering or explanation to ensure relevance reflecting extraneous data.

• While this study marks substantial progress, it would benefit from more rigorous analysis and a deeper interpretation of the findings to strengthen its clinical significance and improve reader comprehension. Additionally, certain sections lack coherence. For instance:

o The introduction presents a limited rationale and requires further expansion to offer a more thorough and well-rounded explanation.

o The results section should go beyond merely presenting data by providing deeper analysis and interpretation, ensuring a logical and seamless flow between findings.

o The clarity and flow, typographical accuracy, and consistency must be part of improvements as essential sections to strengthen the study’s impact and contribution to the field. Consequently, the results demand a more comprehensive analysis and critique to enhance their clarity and relevance.

Reviewer #3: The statastical retrospective review of vulvar cancer patients sheds light to important considerations for women living with HIV for increased monitoring and surveillance to potentially catch vulvar cancers in this population at early disease stages. Minor typographical revisions.

**Do you want your identity to be public for this peer review?** For information about this choice, including consent withdrawal, please see our Privacy Policy

Reviewer #1: **Yes: ** Dr Nadine Rapiti

Reviewer #2: No

Reviewer #3: **Yes: ** Patricia Brothers

---

## [Author Response · Author response to Decision Letter 1]

10 Apr 2025

Reviewers’ comments

Reviewer #1: Minor

1. Include the core diagnosis in the Abstract, Objective and Materials i.e. vulvar cancer

Response: The Abstract has been reviewed to include vulvar cancer

2. Line 45: “Globally, it accounted for over 45 000 new cases and 17 000 deaths” What is the time period referred to?

Response: The statement has been corrected as ‘Globally, it accounted for over 45 000 new cases and 17 000 deaths in the 2020 global cancer estimate with age-standardized incidence rate (ASIR) and mortality of 0.9 and 0.3 respectively’

3. Correct minor grammatical errors throughout: space between age and years, data was …p value lower case in Table 1, spelling

Response: Corrections have been made to address the grammatical errors, spellings and spaces included between age and years, and lower case for p values.

4. Table 1: Is the mean or median tumour size documented? Is the p value for co-morbidity as a group or for DM only?

Response: Yes, the average largest diameter of tumour size is compulsory information that is captured in the oncology files of patients with cervical and vulvar cancer in our institution. The p value was for the group of co-morbidity.

5. In the Discussion of Age, what % of patients in these referenced studies were HIV pos i.e. were similar cohorts compared?

Response: Most of the studies from these countries with an average age at diagnosis above 60 years did not report the proportions of HIV in their cohorts which reflects the low prevalence of HIV to allow for comparison in those regions. However, studies from sub-Saharan Africa with high prevalence of HIV, especially the Southern African countries, reported proportions of HIV in the cohorts of vulvar cancers, ranging between 23% and 89%. We did not discuss the proportions of the cohorts with HIV in the discussion on age to avoid repetition as this was mentioned in the paragraphs that reported on HIV.

Major:

1. Review stats in Table 1eg. HIV neg (69=21.8%) and HIV pos (248=78.2%) for the FIGO stage Early 32 (28.1) vs 82 (71.9) and Advanced 37 (18.3) vs 166 (81.8), with changing the denominator: Early HIV neg 32/69=46.4% vs 82/248=33% in HIV pos and Advanced HIV neg 37/69=53.6% and HIV pos 166/248=66.9%. The change in denominator allows for comparison of HIV pos and HIV neg groups, as the large number of HIV pos cases is a study strength.

Response: Thank you for this suggestion. The entire table has been corrected to include the column outputs rather than the row outputs that were initially captured in the table.

2. Remove replication of results described in words, when already presented in the tables.

Response: We have removed duplication of results in words that were already presented in the tables or charts except where to give more clarity.

Reviewer #2:

The study investigated the clinical, pathological, and treatment characteristics of patients of HIV and non-HIV-related vulvar cancers. However, my concerns/suggestions are not necessarily a reflection of the quality of the work as the following:

Although this may seem like a minor issue, the abstract lacks depth and feels somewhat diluted. It merely touches on the topic without offering substantial insights. The manuscript is submitted as a research article, yet its primary objective aligns with a review perspective. Please refine the abstract and introduction to make it more concise, informative, and aligned with the intended research focus.

Response: The study was aimed to describe the clinical, pathological, and treatment characteristics of patients of HIV and non-HIV-related vulvar cancers by retrospectively reviewing this information. It was not a review article but a clinical audit. We have reviewed the abstract and the introduction sections to give better clarity and conciseness to information being passed by the research.

Some comments about figures/tables here and there, study investigated the clinical, pathological and treatment:

Figure 1:

• The chart could benefit from data labels for individual points, especially for key inflection years like 2016 or 2019.

The notation is somewhat ambiguous; clarifying which trend it applies to would improve interpretability.

• Using exponential for cases and linear for mean age may confuse readers unless justified. A brief explanation of why these particular trends were chosen would add clarity.

• The chart lacks contextual information such as the type of cases being studied, geographical scope, or contributing factors that might explain the trends.

• Consider normalizing the scales for easier visual comparison of the trends.

Response: The chart had been modified to answer the raised queries including data labels. Figure 1a showed the raw data while figure 1b showed the normalized scale.

The trendlines were drawn based on the raw data rather than being adjusted for population size or other demographic standardization. The exponential curve was used as it followed the data distribution more. However, to avoid ambiguity, we have changed the trendlines to linear for both variables and this has been corrected in the result. Also, limited individual data labels were included to avoid clumsiness of notation. A chart showing normalized scale was included to enhance clarity. Geographic or demographic population was not considered in creating the chart. It was based on raw count/frequency of vulvar cancers. There was no identifiable factors for the trend except for the possibility of HIV as explained in the discussion section.

Figure 2:

• The chart does not provide information about the population size or geographic scope. Are these proportions relative to the general population, specific cohorts, or testing populations?

• The y-axis labeled “Proportion of HIV status” is vague. Are the proportions normalized to a specific population size or expressed as percentages?

• The chart does not stratify cases by other variables such as age, gender, or risk factors, which could provide deeper insights.

Response: The chart showed the percentage of vulvar cancer patients who were either HIV positive and negative per year. It was not relative to the general population nor did we stratify the chart based on age but by HIV status. We have reviewed the chart to reflect the comments and suggestions. We included figure 2a which showed the distribution (raw frequency) and figure 2b which showed the normalized scale. The axes labelling has been improved to give more clarity.

Figure 3:

• The chart does not specify whether the frequencies are raw counts or adjusted for population growth, age standardization, or other factors.

• There is no indication of geographic or demographic focus.

• The dual y-axes might confuse some viewers, as it is not immediately clear that each cancer is plotted on a different scale.

Response: Raw counts were used to generate the chart based on the frequency of cases seen per year during the study period. It was not adjusted for population growth, age standardization, percentage annual change, or geographic region. It was based on an institutional data of patients with cervical and vulvar cancers seen during the study period.

We have reviewed the chart and the axes have been labelled appropriately for more clarity. Figure 3a showed the frequency of cervical and vulvar cancer seen per year. The y-axis on the left reflects the vulvar cancer frequency while the y-axis on the right reflects the cervical cancer frequency. Figure 3b showed the normalized scale for the two cancers. The comma between the numbers is a dot in the computer systemin our region i.e. 0,6 is 0.6

Other Comments:

• Their uploaded file, named as Excel sheet for PLoS ONE manuscript Nov 5, 2024, has some areas of discrepancy such as:

The Excel sheet includes several derived variables (e.g., tsize_cat5, age_cat9), which may need further clarification to match the previously provided raw data tables, reflecting the information's depth!

• Some columns, like trend and multiple categorizations (e.g., agecat3, age_cat8, age_cat9), may not directly align with the manuscript’s core tables and could require filtering or explanation to ensure relevance reflecting extraneous data.

Response: The variables such as tsize_cat5, age_cat9, agecat3, age_cat8, age_cat9, follow up, in the Excel sheet for PLos ONE were derived variables from the raw data. They were used to stratify or categorize some of the variables such as age, tumour size and also to identify which variables to use for a particular analysis because of several variable names. Excel was used to generate some figures that did not come right on Stata during the analysis. Hence, the reason while the uploaded data showed some derived variables. The variable “trend” contained rearranged patient numbers ‘1-317’that occurred during analysis, it was not a particular variable. Thank you for the kin and diligent observation. To avoid confusion while maintaining transparency, we have filtered/removed some variables from the minimal data set which did not align with the core results of this manuscripts as suggested while leaving some of the derived variables for ease of result replication as required by PLos ONE for the reviewers.

General comments:

While this study marks substantial progress, it would benefit from more rigorous analysis and a deeper interpretation of the findings to strengthen its clinical significance and improve reader comprehension. Additionally, certain sections lack coherence. For instance:

• The introduction presents a limited rationale and requires further expansion to offer a more thorough and well-rounded explanation.

• The results section should go beyond merely presenting data by providing deeper analysis and interpretation, ensuring a logical and seamless flow between findings.

• The clarity and flow, typographical accuracy, and consistency must be part of improvements as essential sections to strengthen the study’s impact and contribution to the field. Consequently, the results demand a more comprehensive analysis and critique to enhance their clarity and relevance.

Response: Thank you for your comments. We have further done review and analysis to add better interpretation and understanding to the work.

The introduction section has been reviewed and made concise to reflect the aim of the study. The results have been reviewed to limit repetition of data presented in words and in tables or charts. And to also give better interpretation. The grammatical errors have been reviewed to improve coherence and clarity of our findings.

Reviewer #3:

The statistical retrospective review of vulvar cancer patients sheds light to important considerations for women living with HIV for increased monitoring and surveillance to potentially catch vulvar cancers in this population at early disease stages. Minor typographical revisions.

Response: Thank you for the diligent review of our work and for your suggestions. We have reviewed the manuscript for grammatical and typographical errors.

---

## [Decision Letter · Decision Letter 1]

Dear Dr. Sajo,

Thank you for submitting your manuscript to PLOS ONE. After careful consideration, we feel that it has merit but does not fully meet PLOS ONE’s publication criteria as it currently stands. Therefore, we invite you to submit a revised version of the manuscript that addresses the points raised during the review process.

We look forward to receiving your revised manuscript.

Kind regards,

Milad Khorasani, PhD

Academic Editor

PLOS ONE

Journal Requirements:

Reviewers' comments:

Reviewer's Responses to Questions

**Comments to the Author**

Reviewer #1: All comments have been addressed

Reviewer #2: All comments have been addressed

Reviewer #3: All comments have been addressed

2. Is the manuscript technically sound, and do the data support the conclusions?

Reviewer #1: Yes

Reviewer #2: Yes

Reviewer #3: Yes

3. Has the statistical analysis been performed appropriately and rigorously?

Reviewer #1: Yes

Reviewer #2: Yes

Reviewer #3: Yes

4. Have the authors made all data underlying the findings in their manuscript fully available?

Reviewer #1: Yes

Reviewer #2: Yes

Reviewer #3: Yes

5. Is the manuscript presented in an intelligible fashion and written in standard English?

Reviewer #1: Yes

Reviewer #2: Yes

Reviewer #3: Yes

Reviewer #1: The authors have revised the manuscript well.

There are few minor grammatical corrections:

Discussion:

“Aside the population difference” Insert “from”

“Hence, a need for a prospective ongoing study in our institution to describe this impact”. Correct grammar

“The most common vulvar SCC, basaloid and warty types, are mostly driven by high risk HPV infection and has been found to have better prognosis” Correct to singular

Reviewer #2: The authors have successfully addressed all my comments across the manuscript, achieving full resolution. The revised manuscript reflects significant improvements and demonstrates a strong commitment to scientific rigor, clarity, and transparency. It includes additional methodological details, improved statistical and visual reporting, and a more open discussion of limitations. Based on the thoroughness of the revisions and the full resolution of my concerns, I recommend the revised manuscript for acceptance with no further major revisions.

Reviewer #3: (No Response)

**Do you want your identity to be public for this peer review?** For information about this choice, including consent withdrawal, please see our Privacy Policy

Reviewer #1: **Yes: ** Nadine Rapiti

Reviewer #2: No

Reviewer #3: No

---

## [Author Response · Author response to Decision Letter 2]

21 May 2025

Editor’s comments

Response: We critically reviewed the references again and they are complete and none has been retracted. However, reference 35 and 36 were corrected for proper citation.

Reviewers’ comments: Minor grammatical corrections

1. “Aside the population difference” insert from

Response: “from” has been inserted “Aside from the…”

2. “Hence, a need for a prospective ongoing study in our institution to describe this impact”. Correct grammar

Response: “Hence, there is a need for a prospective study in our institution to describe this impact.

3. “The most common vulvar SCC, basaloid and warty types, are mostly driven by high risk HPV infection and has been found to have better prognosis” correct to singular

Response: The most common vulvar SCC, basaloid or warty type, is mostly driven by high risk HPV infection and has been found to have better prognosis.

**** The minimal dataset did not contain any form of identifier. They contained data used to generate results in the manuscript. I did not insert any other data in this revision.

---

## [Decision Letter · Decision Letter 2]

Clinico-pathological and treatment characteristics of HIV and non-HIV related vulvar cancers: Analysis of a South African cohort

PONE-D-24-50571R2

Dear Dr. Sajo,

We’re pleased to inform you that your manuscript has been judged scientifically suitable for publication and will be formally accepted for publication once it meets all outstanding technical requirements.

Kind regards,

Milad Khorasani, PhD

Academic Editor

PLOS ONE

Additional Editor Comments (optional):

Reviewers' comments:

Reviewer's Responses to Questions

**Comments to the Author**

Reviewer #1: All comments have been addressed

2. Is the manuscript technically sound, and do the data support the conclusions?

Reviewer #1: Yes

3. Has the statistical analysis been performed appropriately and rigorously?

Reviewer #1: Yes

4. Have the authors made all data underlying the findings in their manuscript fully available?

Reviewer #1: Yes

5. Is the manuscript presented in an intelligible fashion and written in standard English?

Reviewer #1: Yes

Reviewer #1: (No Response)

**Do you want your identity to be public for this peer review?** For information about this choice, including consent withdrawal, please see our Privacy Policy

Reviewer #1: **Yes: ** Dr Nadine Rapiti

---

## [Editor Report · Acceptance letter]

PONE-D-24-50571R2

PLOS ONE

Dear Dr. Sajo,

I'm pleased to inform you that your manuscript has been deemed suitable for publication in PLOS ONE. Congratulations! Your manuscript is now being handed over to our production team.

Kind regards,

on behalf of

Dr. Milad Khorasani

Academic Editor

PLOS ONE